# Effect of Phasic Experimental Pain Applied during Motor Preparation or Execution on Motor Performance and Adaptation in a Reaching Task: A Randomized Trial

**DOI:** 10.3390/brainsci14090851

**Published:** 2024-08-23

**Authors:** Laïla Badr, Léandre Gagné-Pelletier, Hugo Massé-Alarie, Catherine Mercier

**Affiliations:** 1Centre for Interdisciplinary Research in Rehabilitation and Social Integration (Cirris), Centre Intégré Universitaire de Santé et Services Sociaux de la Capitale-Nationale, Quebec City, QC G1M 2S8, Canada; laila.badr.1@ulaval.ca (L.B.); leandre.gagne-pelletier.1@ulaval.ca (L.G.-P.); hugo.masse-alarie@fmed.ulaval.ca (H.M.-A.); 2School of Rehabilitation Sciences, Laval University, Quebec City, QC G1V 0A6, Canada

**Keywords:** motor control, motor learning, upper limb, nociceptive stimuli, laser stimulation, force field adaptation, sensorimotor integration, learning interference

## Abstract

Musculoskeletal conditions often involve pain related to specific movements. However, most studies on the impact of experimental pain on motor performance and learning have used tonic pain models. This study aimed to evaluate the effect of experimental phasic pain during the preparation or execution of a reaching task on the acquisition and retention of sensorimotor adaptation. Participants were divided into three groups: no pain, pain during motor preparation, and pain during motor execution. Pain was induced over the scapula with a laser while participants performed a force field adaptation task over two days. To assess the effect of pain on motor performance, two baseline conditions (with or without pain) involving unperturbed pointing movements were also conducted. The results indicated that the timing of the nociceptive stimulus differently affected baseline movement performance. Pain during motor preparation shortened reaction time, while pain during movement execution decreased task performance. However, when these baseline effects were accounted for, no impact of pain on motor adaptation or retention was observed. All groups showed significant improvements in all motor variables for both adaptation and retention. In conclusion, while acute phasic pain during motor preparation or execution can affect the movement itself, it does not interfere with motor acquisition or retention during a motor adaptation task.

## 1. Introduction

Training, aiming at learning new motor tasks, often occurs in the context of pain, such as during physical rehabilitation after trauma or when an athlete pursues competition despite an injury. Some studies conducted on both animals and humans suggest that pain might interfere with motor learning, but the evidence remains highly conflicting [1,2,3,4]. Several factors may contribute to these discrepancies, including the type of motor learning (e.g., motor sequence learning, motor adaptation, visuomotor learning), whether acquisition or retention was assessed (and whether retention was tested in the presence or absence of pain), the effector involved in the task (e.g., trunk, arm, leg, tongue), and the type of pain (e.g., experimental vs. clinical) [1].

A recent systematic review indicated that all studies using experimental pain in humans employed tonic pain models [1], which involve long-lasting pain not modulated by movement. Tonic pain modalities that were employed included capsaicin [5,6,7,8,9,10,11,12], hypertonic saline injection [13,14,15,16], or thermode [17]. In contrast, phasic pain is short lasting and can be triggered in relation to specific movements. Common phasic pain models include laser and electrical stimulation [18]. Studying the impact of phasic pain on motor learning is clinically relevant because pain is often evoked by movements in musculoskeletal disorders [19], a phenomenon named movement-evoked pain. Movement-evoked pain is being increasingly studied and was found to be associated with long-term disability in older adults with chronic low back pain [20]. Therefore, the lack of studies investigating the impact of phasic and movement-evoked pain constitutes an important gap that needs to be addressed. Additionally, this experimental pain model allows us to provide insights into the mechanisms underlying pain’s effect on motor learning. Indeed, the effect of phasic pain on corticospinal excitability is very limited in time (with inhibition shown at an interstimulus interval of approximately 200 ms when laser stimulation is applied to the hand [18]), which allows us to test how interfering with specific phases of movement preparation or execution affects motor learning. Being able to determine whether the specific timing of pain in relation to movement impacts interference with motor learning would fill another research gap.

In animals, transient interference with motor cortex (M1) activity (via brief intracortical microstimulation) during the motor preparation phase impairs motor acquisition [21]. In humans, although much behavioral and neurophysiological evidence highlights the importance of motor preparation in voluntary movements [22], few studies have focused on its role in motor learning. For example, the motor preparation phase is necessary to improve motor performance during an adaptation task in a force field. Without this phase, the motor execution phase alone does not allow participants to improve their motor performance [23]. Additionally, improvement in motor performance during training is associated with a decrease in M1 inhibition in the late phase of motor preparation [24] and a decrease in the amplitude of early and late components of the contingent negative variation (CNV) [25], a characteristic brain wave of motor preparation [26]. These studies suggest the importance of motor preparation for motor learning, though they do not establish a causal relationship. Thus, an interference of pain during motor preparation could negatively influence motor learning.

One might argue that pain occurring during movement execution could have a lesser impact on motor learning if it does not perturb the movement itself. However, it has been shown that movement-related pain can modulate corticospinal excitability during motor preparation, i.e., before the occurrence of nociceptive stimuli [27]. Anticipation of pain has also been shown to affect movement initiation (reaction time) and speed and reduce corticospinal excitability [27,28,29,30]. Cognitive factors also play a role; for instance, whether the timing of pain is predictable has been shown to modulate pain-induced changes in corticospinal excitability [31]. This suggests that pain occurring during movement execution might not only impact task performance but might also interfere with motor preparation. To distinguish between different impacts that pain might have on motor performance and adaptation, it is of interest to consider variables that reflect different aspects of motor control. Reaching movements can be broadly separated into two components: 1, feedforward control involved in initiating movement, and 2, feedback control to produce corrective movements (although these processes are not necessarily completely segregated) [32,33].

The objective of this study was to evaluate the effect of experimental phasic pain occurring during the preparation or execution of a reaching task on the acquisition and retention of sensorimotor adaptation. To perform this, three groups of participants (no pain; pain during motor preparation; pain during motor execution) performed a force field adaptation task on two consecutive days (Day 1, acquisition; Day 2, retention) in a robotic exoskeleton system. To characterize the effect of pain on motor performance, two baseline conditions (with or without pain for the two groups receiving pain) involving unperturbed reaching movements were performed and compared. Interference effects on learning were measured via performance during acquisition (by comparing early vs. late performance on Day 1) and 24 h retention (by comparing early performance on Day 1 vs. Day 2), as immediate performance improvements during acquisition do not necessarily represent long-term effects [34]. It was hypothesized that phasic pain occurring during the motor preparation phase would interfere with both motor acquisition and retention and, in particular, on variables reflecting feedforward control. It was expected that phasic pain occurring during movement execution would cause less interference or would interfere with task performance rather than feedforward processes.

## 2. Materials and Methods

### 2.1. Sample

Healthy individuals aged between 18 and 50 years old were included. Exclusion criteria were 1, the presence of recent (<6 months) musculoskeletal problems involving the upper limb (including chronic or acute pain); 2, an antecedent of neurologic psychiatric illness; 3, the presence of a skin condition (e.g., psoriasis, urticaria); and 4, an uncorrected visual impairment. A sample size of 16 participants by group was selected based on previous studies investigating the effect of experimental pain on motor learning (median of 12, range between 9 and 20 [1]), given that it was the first study employing this specific motor learning task. Participants were recruited over a period from February 2023 to February 2024 and were assigned to one of three groups using a randomization sequence generated in Excel, ensuring an equal allocation ratio of 1:1:1. There were no specific restrictions or blocking used in the randomization process. Participants were assigned to the groups sequentially based on their order of enrollment until the targeted sample size was reached in each group. Blinding was not possible given the nature of the independent variable. The first two groups received nociceptive stimuli during either the preparation (preparation group) or the execution (execution group) of the motor task, while the third did not receive any nociceptive stimulation (control group).

### 2.2. Study Design

Each participant took part in two experimental sessions carried out 24 h apart at CIUSSS de la Capitale-Nationale. On both days, a reaching task was performed with the right upper limb using a Kinarm Exoskeleton Lab (see Section 2.3 below).

On Day 1, participants performed three blocks of the reaching task: two baseline and one adaptation block (see Figure 1). Two baseline blocks were performed to assess whether the pain modified their initial motor performance, i.e., in the two pain groups, the first block was performed pain free, while pain was applied during the second block. Although they received no nociceptive stimuli, the control group performed two baseline blocks as well to receive the same amount of practice. Immediately after the last trial of the second baseline block, a force field unexpectedly started to be applied (adaptation block). Participants were informed that a perturbation was going to be applied at some point in the experiment, but the timing and nature of this perturbation were unknown. The instructions given to the participant were to come back as much as possible to their baseline movement pattern.

On Day 2, the participants performed only one block with force field exposure (retention block, which was like the adaptation block on Day 1). No washout period was performed at the end of Day 1, and no force-field-free baseline measurements were performed at the beginning of Day 2, as this might have reduced retention.

The two pain groups received nociceptive stimuli during Baseline 2, adaptation (Day 1), and retention (Day 2) blocks, but the timing of application differed (see Section 2.4 below). During the experiment, participants from each group had to verbally rate the intensity of their pain before the first block, between the blocks, and at the end of the last block using a numerical rating scale ranging from 0 (no pain) to 10 (worst pain imaginable).

### 2.3. Instrumentation

Kinarm: The Kinarm Exoskeleton Lab (Kinarm, Kingston, ON, Canada) is a robotized exoskeleton allowing combined movements of the shoulder and the elbow joints to move the upper limb in the horizontal plane. It includes a 2D virtual environment allowing us to present visual stimuli with an appropriate perception of depth. It allows us to accurately measure the spatial and temporal parameters of movement and apply controlled mechanical perturbations. Elbow and shoulder joint angular positions were obtained from Kinarm motor encoders and sampled at 1 kHz. The position of the index finger is calculated in real time by the Dexterit-E software (version 3.9.4) of the Kinarm system. Data processing was performed with Matlab (MathWorks, R2022a).

Laser Nd:YAP: Phasic experimental pain was applied with an Nd:YAP laser (Stimul 1340, Deka, Florence, Italy), which allows us to selectively activate the nociceptive fibers Aδ and C without concomitant activation of the mechanoreceptors or the Aβ fibers [35,36]. During the experimental paradigm, stimuli were applied to the right shoulder blade. To avoid any phenomenon of habituation or sensitization, the laser irradiation zone was slightly moved by the experimenter between trials following a grid of points spaced one centimeter apart drawn on the participant’s skin [36]. Before starting the experiment, a few laser stimuli (~10) were applied to the participant’s left shoulder blade. This was performed to establish the laser intensity threshold at which the participant rated the pain between 3 and 4 on a scale of 10 and to familiarize him with the experience of experimentally induced pain. The laser parameters were as follows: spot size: 7 mm; duration: 5 ms; and intensity: 3.5 J to 4.75 J (depending on individual calibration).

### 2.4. Task Description

Subjects performed ballistic right arm reaching movements towards one of the two visual targets (radius of 1 cm; presented in an 80/20 random sequence) located 10 cm from the central starting position: a FAR and a NEAR target—requiring multi-joint coordination between the elbow (flexion–extension) and the shoulder (adduction–horizontal abduction). An instructed delay reaction time (RT) task with two movement directions specified using a visual cue was selected for several reasons. First, it allowed us to control the duration of the motor preparation period and when the noxious stimulus occurred in relation to motor preparation processes, which is important given the fact that corticospinal excitability is dynamically modulated during the motor preparation period [37]. Second, it has been shown that applying a nociceptive laser during movement execution in a specific direction but not in the other results in the acquisition of movement-related fear, supporting the ecological relevance of such a paradigm [27,38]. Third, reaching toward two different targets, with only one involving a force field perturbation, aimed to slow down adaptation, which tends to occur very fast in this type of task [11,39]. The two targets were at an equal distance from the starting position of the index (based on the following starting position of the right arm: 50° of horizontal abduction at the shoulder and 90° at the elbow). Subjects were instructed to “shoot” through the target as quickly and precisely as possible as soon as it appeared. The force field and noxious stimulations were applied only for the FAR target, which was the target of interest and was present in 80% of the trials.

Each trial proceeded as follows (see Figure 2):(1)Starting position: the participant holds his index finger (represented by a white dot) in the starting target for 2000 ms;(2)Motor preparation phase (duration = 500 ms): the target to be reached (NEAR or FAR) appears in red (preparatory signal), and the participant is instructed to maintain his index finger in the starting target as long as the target is red;(3)The target to be reached switches to green and the participant initiates a reaching movement;(4)Once the target is reached or when the subject crosses the 10 cm virtual radius (i.e., misses the target), a dampening field is applied to rapidly stop the movement. The robot then passively returned the hand to the starting position;(5)The participant must maintain his index at the starting position for 1000 ms before moving on to the next trial.

A total of 115 trials (92 toward the FAR target) were conducted on Day 1 and 65 trials (52 toward the FAR target) on Day 2. Only trials toward the FAR target were retained for analysis.

Force field application: For the adaptation (Day 1) and retention (Day 2) blocks, the robot applied a velocity-dependent resistance force (−4 Nm·s/rad) at the elbow during trials toward the FAR target. Thus, the participant had to adapt his behavior (sensorimotor adaptation) to successfully attain the task goal.

Nociceptive stimuli application: For the preparation group, the noxious stimuli were applied during the motor preparation phase, i.e., in a window between 450 and 350 ms before the start signal (GO, Figure 2). This timing aligns with the typical duration of the motor preparation phase preceding voluntary movements, which is 500 ms before movement onset [40]. As we estimated the shortest delay for pain perception to be ~300 ms following the depolarization of Aδ nerve fibers by the laser [41], pain perception occurred late during the motor preparation phase (~50 to 150 ms prior to the GO), corresponding to the moment at which modulation of corticospinal excitability has been shown to be maximal [37]. For the execution group, nociceptive stimuli were applied as soon as the subject exited the starting position after the appearance of the GO signal, causing pain perception to occur toward the maximal movement amplitude, as the average movement duration was 350.8 ms.

Visual feedback on performance: At the end of each trial, the target’s color provided information about the accuracy and time required to cross the target. For a trial to be considered successful, the duration of the movement had to be less than 700 ms (including the RT). If the movement duration was less than 700 ms and the index finger hit the target, the target turned blue. If the movement duration was less than 700 ms but the index finger missed the target, or if the movement duration was too long, the target disappeared. This visual feedback was important, especially with respect to the speed of movement, to obtain ballistic movements with similar speeds (and, therefore, similar force field amplitudes) in all conditions and subjects. Note that although participants were told that it was required to maintain a similar movement speed across the experiment, they were not explicitly informed that the amplitude of the force field was dependent on movement speed.

### 2.5. Variables

Four variables were used to assess motor performance: the initial angle of deviation (iANG), the final error (fERR), the reaction time (RT), and the movement time (MT) [39].

The iANG is defined as the angle between two vectors: 1, one going from the starting position to the target to be reached, and 2, one going from the index starting position to the one corresponding to the first acceleration peak. It reflects the motor planning of the participant, as it is based only on the initial part of the movement and, therefore, is unlikely to be influenced by sensory feedback (see Figure 2b). The fERR was used as a task performance indicator because the explicit goal of the task was to cross the target as accurately as possible. It was measured as the angle between two vectors: 1, one between the index starting position and the target to be reached, and 2, one between the index starting position and its position when it crossed the invisible 10 cm radius circle (see Figure 2b) [11,37]. It is important to note that the iANG is a signed value, with negative values of the iANG reflecting under-compensation of the force field (i.e., deviation in the direction the force field pushes the limb). In contrast, positive values reflect over-compensation of the force field. In contrast, an absolute value was used the fERR. Given the size of the target and its distance, an fERR > 5° indicates that the target was missed.

The RT is defined as the time elapsed between the GO signal and the initiation of the movement, i.e., the moment at which the index leaves the starting position.

The MT corresponds to the time elapsed between the initiation of the movement and the end of the trial, corresponding to the moment where the index crosses the 10 cm invisible radius. As the force field applied was velocity dependent, the MT was calculated to verify that a comparable force field perturbation was applied across groups and days.

For each variable, the median (to limit the impact of extreme trials) of the trials in specific periods of interest in each block was extracted as follows:Baselines 1 and 2: last 10 trials of each baseline block;Early adaptation: first 5 trials of the adaptation block;Late adaptation: last 10 trials of the adaptation block;Early retention: first 5 trials of the retention block.

Note that a lower number of trials were included in the period of interest during the early stages of exposure to the force field as motor adaptation occurs rapidly.

### 2.6. Statistical Analyses

Descriptive statistics are reported as mean ± standard deviation. The normality of data distributions was verified using the Shapiro–Wilk test. The homogeneity of variances was verified, and a Huynh–Feldt correction was applied if the sphericity assumption was violated.

*T*-tests were performed to compare sociodemographic characteristics, laser intensity, and pain ratings across the three groups.

Two-way repeated measures analyses of variance (ANOVAs) were performed to compare the three groups (either receiving pain (preparation and execution groups or not (control group)) across different time points as follows:Between Baseline 1 and Baseline 2 (Day 1 only): to determine whether the presence of pain impacted baseline motor performance;Between early and late adaptation (Day 1 only): to determine the effect of the group (i.e., the presence of pain and its timing) on motor acquisition;Between early adaptation (Day 1) and early retention (Day 2): to determine the effect of the group (i.e., the presence of pain and its timing) on motor retention.

Post hoc analyses were performed using Tukey’s correction for multiple comparisons.

All these pre-specified statistical analyses were conducted using Prism 10 software (GraphPad Software, Boston, MA, USA). The alpha threshold was set at 0.05 for all analyses. No formal interim analyses were performed during this study.

## 3. Results

### 3.1. Sample Description

Sixty-two participants were initially included, but forty-eight healthy individuals (twenty-eight females/twenty males) completed this study. Fourteen subjects were excluded after their first visit as they were unable to comply with task requirements (too many false starts, defined as a reaction time < 100 ms; see Figure 3 for participant flow). The sociodemographic characteristics of the three groups are presented in Table 1 (reflecting only the participants who completed this study and were included in the analyses (n = 48 for all analyses)), as well as information on nociceptive stimulation and pain ratings.

### 3.2. Time Course of the Different Variables across the Experiment

Figure 4 shows the time course for each of the four variables of interest in the control group to illustrate the behavior expected in this type of motor adaptation task. Large deviations of the iANG and increases in the fERR are seen at the beginning of the adaptation period, corresponding to the moment at which participants are first exposed to the mechanical perturbation. Participants then adapted to the perturbation, coming back to a motor performance that is closer to the one observed during baseline, i.e., prior to the exposure to the perturbation. On Day 1, the force field perturbation induced substantial movement errors during early adaptation. However, participants rapidly adapted their motor strategy (as shown by an increase in the iANG during the adaptation block in Figure 4a), resulting in an improvement in task performance (as seen by a decrease in the fERR during the adaptation block in Figure 4b). On Day 2, both variables remained at the same level as during Day 1 late adaptation, supporting the retention of motor acquisition (see Figure 4a,b). The RT showed a decrease over time (see Figure 4c). The MT showed a transient increase during adaptation but came back to baseline values at the end of adaptation and during retention (see Figure 4d). Statistical comparisons between groups at the different time points are presented in the next sections and synthesized in Table 2.

### 3.3. Effect of Pain on Baseline Motor Performance

The ANOVA performed to evaluate the influence of pain on baseline motor performance indicated an effect of Group (*p* = 0.012) but no effect of Time (*p* = 0.488) or Time × Group interaction (*p* = 0.434) on the iANG. Post hoc analysis demonstrated that the preparation group differed from the two others, with iANG values that were closer to 0 (see Figure 5a). However, given that no interaction was observed, this difference is not explained by the presence of pain.

For the fERR, the ANOVA revealed a significant Time × Group interaction (*p* = 0.018), with post hoc analysis revealing that pain induction between the two baselines adversely affected motor performance for the execution group only (see increased fERR in Figure 5b; *p* < 0.0004).

A significant Time × Group interaction (*p* = 0.017) was also observed on the RT, which was explained by a significant decrease in the RT between Baseline 1 and 2 for the preparation group only (i.e., movement was initiated earlier when pain was applied during motor preparation; see Figure 5c). However, no significant effect of Time (*p* = 0.063), Group (*p* = 0.894), or Time × Group interaction (*p* = 0.106) was observed on the MT (see Figure 5d).

Because of the observed effects of pain on baseline motor performance, data (iANG, fERR, RT, and MT) from all time periods during adaptation and retention were normalized against Baseline 2 performance (i.e., the value of Baseline 2 was subtracted from all other time points). Normalized values were then used for adaptation and retention analyses.

### 3.4. Effect of Pain on Adaptation

For the iANG, the adaptation of the motor strategy was evidenced by a main effect of Time (*p* < 0.0001), but no significant effect of Group (*p* = 0.179) or Time × Group interaction (*p* = 0.873) was observed (see Figure 6a).

For the fERR, an improvement in task performance was evidenced by a main effect of Time (*p* < 0.0001), but no significant effect of Group (*p* = 0.651) or Time × Group interaction (*p* = 0.290) was observed (see Figure 6b).

A significant effect of Time on the RT (*p* < 0.0001) was also found, indicating faster movement initiation at late compared to early adaptation, but no significant effect of Group (*p* = 0.600) or Time × Group interaction (*p* = 0.216) was observed (see Figure 6c).

A significant effect of Time on the MT (*p* < 0.0001) was also found, indicating faster movement at late compared to early adaptation), but no significant effect of Group (*p* = 0.651) or Time × Group interaction (*p* = 0.290) was observed (see Figure 6d). The absence of difference between groups on the MT is methodologically important given that the perturbation was velocity dependent.

### 3.5. Effect of Pain on Retention

For the iANG, the next-day retention of the motor strategy was demonstrated by a significant difference between Day 1 early adaptation and Day 2 early retention (*p* < 0.0001). There was no significant Group effect (*p* = 0.522) and no significant Time × Group interaction (*p* = 0.290), indicating that all three groups evolved over time and adjusted in anticipation of the force field perturbation from the beginning of the retention block on Day 2 (see Figure 7a).

For the fERR, retention was demonstrated by a significant difference between Day 1 early adaptation and Day 2 early retention (*p* < 0.0001). There was no significant Group effect (*p* = 0.775) and no significant Time × Group interaction (*p* = 0.899), indicating that all three groups decreased their error and exhibited retention on Day 2 (see Figure 7b).

A significant effect of time on the RT (*p* < 0.0004) was found, indicating faster movement initiation on Day 2 early retention compared to Day 1 early adaptation, but no significant effect of Group (*p* = 0.834) or Time × Group interaction (*p* = 0.707) was observed (see Figure 7c).

A significant effect of Time on the MT (*p* < 0.0001) was also found, indicating faster movement at early retention compared to early adaptation, but no significant effect of Group (*p* = 0.483) or Time × Group interaction (*p* = 0.828) was observed (see Figure 7d).

## 4. Discussion

The results of this study show that acute phasic pain impacted movement performance differently according to the timing of the application of the nociceptive stimulus. Pain occurring during the motor preparation period shortened the RT, without having an impact on the purely feedforward component of movement (iANG), on task performance (fERR) or speed (MT). However, pain occurring during movement execution decreased task performance (causing larger fERR) without impacting the RT, MT, or iANG. However, when these baseline effects were accounted for, no impact of pain on motor adaptation or retention was observed. All three groups showed significant improvements in the iANG, fERR, MT, and RT both from early Day 1 to late Day 1 (adaptation) and from early Day 1 to early Day 2 (retention).

This study was the first to focus on the effect of phasic pain on motor adaptation and retention, but the results are in line with the conclusions of two recent reviews reporting inconsistent (or at least limited) impacts of acute experimental pain on motor learning [1,4]. Importantly, previous studies using a motor adaptation task (i.e., adaptation to a force field perturbation) with a tonic pain model reported limited impact on the task performance measures (corresponding to fERR in the present study) but rather changes in motor strategy [9,11,15,16]. Among these studies, those focusing on adaption during gait showed a switch to a less feedforward strategy [9,15,16], while the only study focusing on adaptation during reaching showed a more feedforward strategy [11]. In the present study, no difference was observed across groups in the purely feedforward component of movement (iANG), suggesting that phasic pain did not have an effect similar to that of tonic pain on motor strategies. Of course, the effect of phasic pain during movement might critically depend on its specific timing in relation to movement, which motivated the choice of comparing two different timings of application. Although less ecologically relevant, pain occurring during movement preparation was hypothesized to have an impact on motor acquisition given the fact that transient interference with M1 activity during the motor preparation phase has been shown to impair motor acquisition in an animal study [21]. Observing an effect of pain during movement execution on the iANG was less likely, as pain was occurring after the feedforward component of movement, but remained plausible given that the anticipation of pain has been shown to modulate corticospinal excitability during motor preparation [27,30].

Two other important factors limiting the generalization of the results need to be considered in relation to the pain model that was employed in the present study. First, pain was applied at the level of the scapula, while most previous studies had more distal application sites. This site was selected due to methodological constraints. Indeed, it was impossible to apply nociceptive stimuli with a laser on more distal locations because of the rapid movements occurring and the robotic system configuration. The only option to apply nociceptive stimuli distally during movement would have been electrical stimulation, which would not have allowed us to selectively activate nociceptors. The scapula was selected as a biologically plausible site where someone could experience pain while performing a reaching movement involving the shoulder. However, much fewer data are available on the neurophysiological effect of pain on trunk muscles [18,42], and the attentional effect of pain, which has been discussed as a potential contributing factor [4,6], might also be different, depending on whether pain is applied on the trunk vs. the limb. Second, pain intensity was rated quite low (average of 2/10 across pain groups and time points), even though nociceptive stimuli were initially calibrated to produce a pain intensity between 3 and 4 out of 10 (which has been shown to generate a significantly larger decrease in corticospinal excitability compared to pain intensity around 1/10) [31]). These lower ratings during the task might be explained by the fact that movement (and even movement preparation or observation) can result in the gating of sensory information [43,44,45,46,47,48,49,50,51,52,53]. Despite these limitations, it is important to keep in mind that pain applied both during the motor preparation and during the execution of movement had an impact on the movement itself (as shown by differences between the two baseline measurements), although they did not impact motor learning. Using higher stimulation intensities would have been likely to result in larger movement perturbations, making it difficult to distinguish between the effect of the pain itself and the effect of the motor perturbation resulting from the stimulation. Another potential limitation of this study is the relatively small sample size (although comparable to that of previous studies in the field), which limits the ability to take the potential effects of sex and gender into account. Sex and gender have been shown to be associated with differences in acute pain sensitivity, although these effects remain controversial and might vary between types of stimuli [54]. In the present study, this effect might have been minimized by the fact that the nociceptive stimuli were calibrated individually. Post hoc exploratory analyses showed no difference between males and females in either the laser intensity used or in the ratings of pain intensity (all *p* > 0.156). Musculoskeletal morphology and strength also differ between males and females [55,56], which might have impacted the ability to adapt to the force field (which was similar across all participants, irrespective of their strength). Although this might have increased within-group variability, this is unlikely to have biased the results given that the male/female ratio did not differ between groups.

Although the pain model used is different, our observation that phasic pain during motor preparation shortened the RT is consistent with previous studies showing that ongoing pain shortens the RT without affecting the accuracy of movement [28,57]. This shortening in the RT was found to be associated with pain-related suppression of beta oscillations over contralateral premotor areas [28]. Using a different paradigm, painful (but also nonpainful) thermal stimuli have been shown to reduce readiness potentials [58]. However, the functional relevance of this effect on movement-preparatory brain activity remains unknown. An alternative explanation for the shortening of the RT could be related to the fact that the nociceptive stimuli occurred close to the visual go signal. Indeed, it has been shown that in a simple RT task, subjects respond faster to simultaneous visual and tactile stimuli than to single visual or tactile stimuli [59]. The fact that no change in the RT was observed when pain occurred during movement execution contrasts with former studies showing that anticipation of movement-related pain affects the RT [27] as well as readiness potentials [60]. A potential explanation is that these former studies included more trials with a stable and simple motor requirement, while in the present study, subjects had to learn a new motor task. The need to focus on a difficult motor task might have limited the effect of pain conditioning.

## 5. Conclusions

In conclusion, this study showed that while acute phasic pain occurring in healthy subjects during motor preparation or execution can impact the movement itself, it does not interfere with motor acquisition or retention during a motor adaptation reaching task. Nevertheless, given the substantial heterogeneity observed in studies focusing on the effect of pain on motor learning and the variety of motor learning paradigms, it would be of interest to further investigate the effects of phasic pain on movement and motor learning. Using a more distal motor task with local application of nociceptive stimuli of higher intensity would address some of the limitations of the current study. Another interesting avenue would be to investigate whether phasic pain has a greater impact if its intensity is modulated in real time as a function of task performance. For instance, the intensity of the nociceptive stimuli could co-vary with the increase in EMG amplitude occurring to counteract the perturbation. Although methodologically challenging, this approach would present a high ecological validity.

## Figures and Tables

**Figure 1 brainsci-14-00851-f001:**
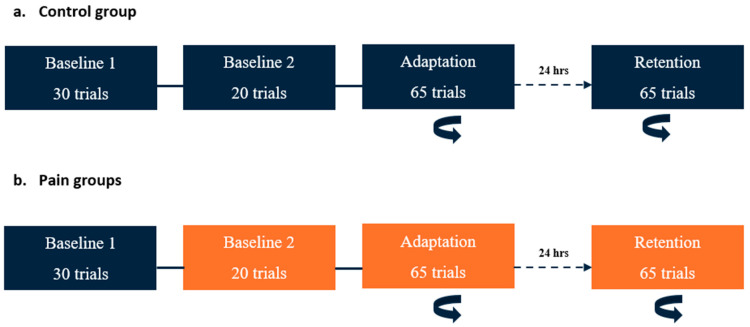
Time course of the experimental paradigm. The rotating arrow indicates the blocks in which a perturbation (velocity-dependent force field) was applied at the elbow. Blocks in orange are those during which nociceptive stimuli were applied. (**a**) Control group: On Day 1, two baseline blocks (without force field) were conducted. The adaptation period (with force field) began immediately after the completion of Baseline 2. On Day 2 (24 h later), a single block with perturbation (retention) was performed. (**b**) Pain groups: The timeline was the same as for the control group, except that nociceptive stimuli were applied during Baseline 2, adaptation, and retention blocks for both pain groups (preparation and execution).

**Figure 2 brainsci-14-00851-f002:**
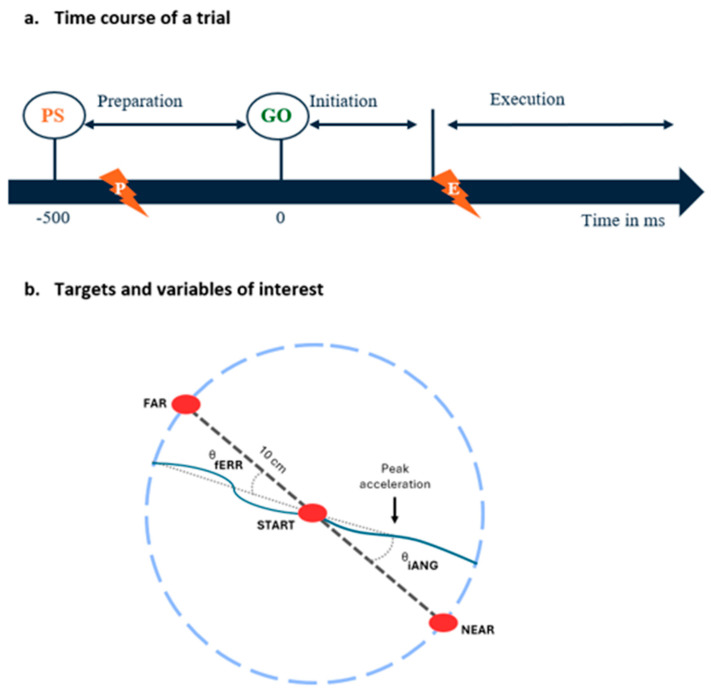
(**a**) Time course of a trial: At each trial, subjects had to reach toward one of two visual targets presented in an 80 (FAR)/20 (NEAR) random sequence. The target to be reached was indicated by a preparatory signal (PS) presented 500 ms prior to the GO signal. Nociceptive stimuli were applied during the motor preparation phase (preparation group, indicated by the lightning with a P) or during the motor execution phase (execution group, indicated by the lightning with an E). (**b**) Targets and variables of interest: Location of the starting position, the FAR and NEAR targets, and an illustration of the kinematic variables extracted from index finger trajectories using examples of typical trials early in the adaptation period (i.e., when trajectories strongly deviate in the direction of the force field compared to the vector between the starting position and the target, indicated by the dashed line). The vector between the index starting position and the first acceleration peak and the vector between index starting position and its position when it crossed the invisible 10 cm radius circle are represented by dotted lines, and were respectively used to calculate the initial angle of deviation (iANG) and the final error (fERR).

**Figure 3 brainsci-14-00851-f003:**
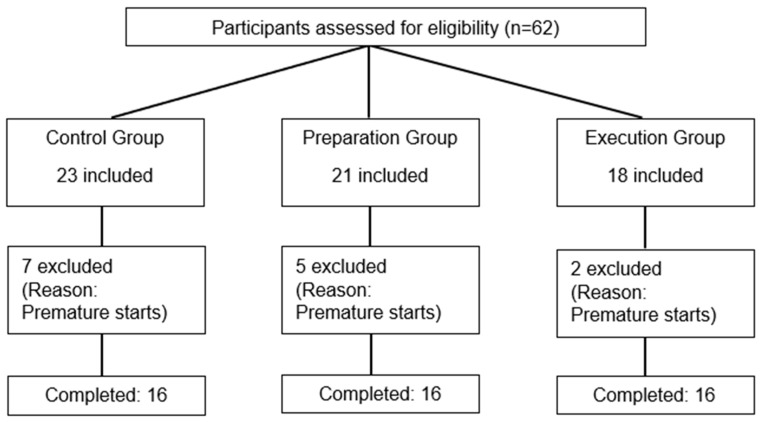
Flow diagram of participant recruitment, assignment, and exclusion.

**Figure 4 brainsci-14-00851-f004:**
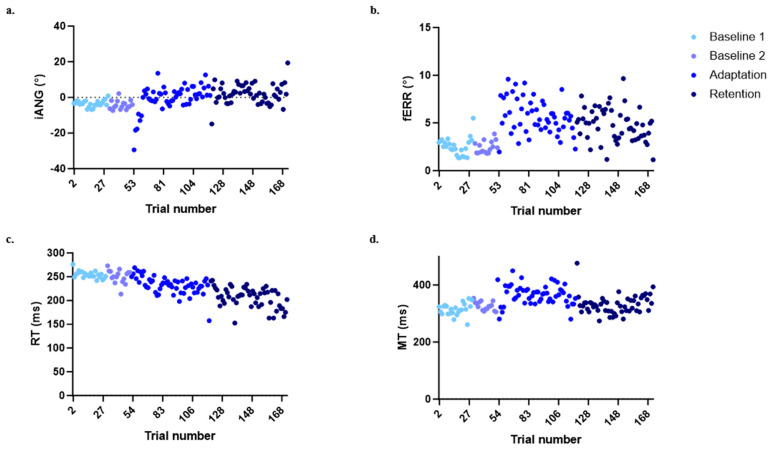
Evolution in the four variables of interest over the time course of the experiment for the control group. (**a**) Initial deviation angle (iANG). (**b**) Final error (fERR). (**c**) Reaction time (RT). (**d**) Movement time (MT)). Each point represents the median of all participants for a given trial.

**Figure 5 brainsci-14-00851-f005:**
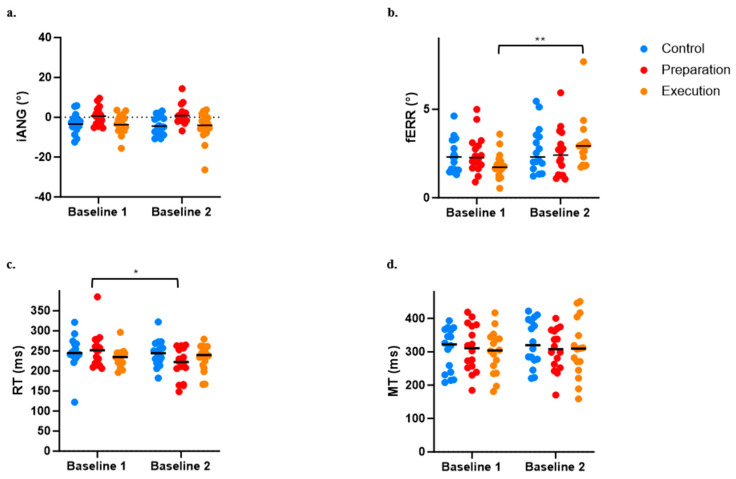
Comparison between Baseline 1 and Baseline 2 across the three groups for the four variables of interest. (**a**) Initial deviation angle (iANG). (**b**) Final error (fERR). (**c**) Reaction time (RT). (**d**) Movement time (MT). * *p* < 0.05; ** *p* < 0.001.

**Figure 6 brainsci-14-00851-f006:**
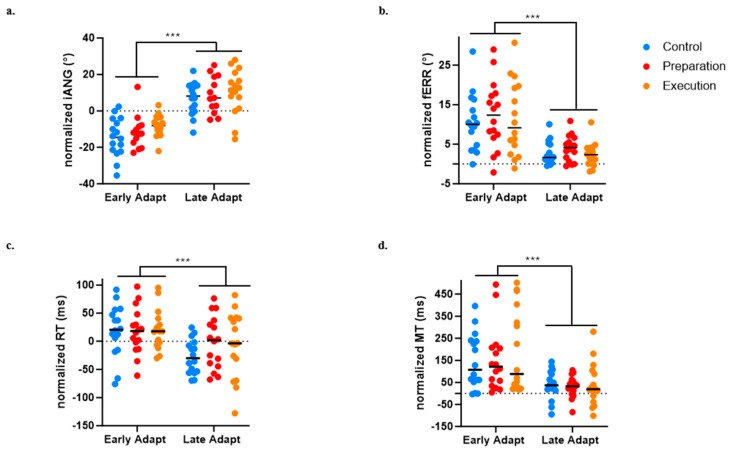
Comparison between early and late adaptation of Day 1 across the three groups for the four variables of interest. (**a**) Initial deviation angle (iANG). (**b**) Final error (fERR). (**c**) Reaction time (RT). (**d**) Movement time (MT)). *** *p* < 0.0001.

**Figure 7 brainsci-14-00851-f007:**
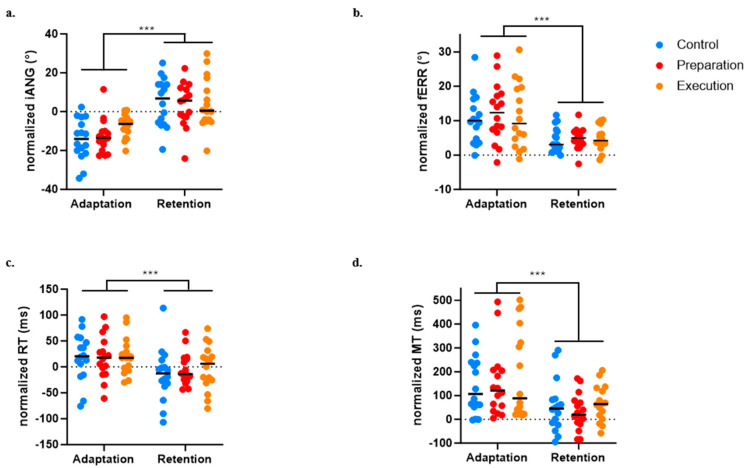
Comparison between early adaptation (Day 1) and early retention (Day 2) across the three groups for the four variables of interest. (**a**) Initial deviation angle (iANG). (**b**) Final error (fERR). (**c**) Reaction time (RT). (**d**) Movement time (MT). *** *p* < 0.0001.

**Table 1 brainsci-14-00851-t001:** Sociodemographic characteristics, laser intensity, and pain ratings for each group.

	Control N = 16	Preparation N = 16	Execution N = 16	*p*-Value
**Sociodemographic characteristics**				
Age, years (mean ± SD)	25.5 ± 4.50	25.6 ± 5.03	24.0 ± 4.50	0.54
Females (n (%))	10 (62.50)	9 (56.25)	9 (56.25)	0.91
Right handed (n (%))	16 (100)	15 (93.75)	16 (100)	0.36
**Laser intensity**, J (mean ± SD)	N/A	4.00 ± 0.32	4.08 ± 0.33	0.50 ^1^
**Pain ratings** (/10; mean ± SD)				
Day 1:				
Before Baseline 1	0	0	0	
End of Baseline 1	0	0	0	
End of Baseline 2	0	2.20 ± 1.07	1.66 ± 1.17	0.48 ^1^
End of Adaptation	0	2.23 ± 1.21	1.91 ± 1.51	0.82 ^1^
Day 2:				
Before Retention	0	0	0	
End of Retention	0	2.06 ± 0.94	1.97 ± 1.18	0.99 ^1^

J: Joules; SD: standard deviation; N/A: non applicable. ^1^ For these variables, *p*-values reflect only the comparison between both pain groups.

**Table 2 brainsci-14-00851-t002:** Mean, standard deviation, and *p*-values for the four variables of interest: initial deviation angle (iANG), final error (fERR), reaction time (RT), and movement time (MT). Note that values reported are raw values for Baselines 1 and 2 and normalized values for the other time points. Statistically significant results are highlighted in bold.

		Period 1 (Mean ± SD)	Period 2 (Mean ± SD)	Time (*p*-Value)	Group (*p*-Value)	Interaction (*p*-Value)
**iANG (°)**						
Baseline 1 vs. Baseline 2	Control	−3.2 ± 5.1	−3.7 ± 4.8	0.488	**0.012**	0.434
	Preparation	0.7 ± 4.6	1.3 ± 4.9			
	Execution	−3.6 ± 5.0	−4.7 ± 7.6			
Early Adaptation vs. Late Adaptation	Control	−14.3 ± 10.7	7.3 ± 8.6	**<0.0001**	0.179	0.873
	Preparation	−12.6 ± 8.7	7.8 ± 10.9			
	Execution	−7.6 ± 5.9	11.1 ± 12.9			
Early Adaptation vs. Early Retention	Control	−14.3 ± 10.65	4.9 ± 12.5	**<0.0001**	0.522	0.290
	Preparation	−12.6 ± 8.65	3.6 ± 11.3			
	Execution	−7.6 ± 5.91	4.4 ± 13.2			
**Ferr (°)**						
Baseline 1 vs. Baseline 2	Control	2.4 ± 1.0	2.7 ± 1.3	**<0.0004**	0.928	**0.018**
	Preparation	2.5 ± 1.1	2.5 ± 1.3			
	Execution	1.8 ± 0.7	3.0 ± 1.4			
Early Adaptation vs. Late Adaptation	Control	10.5 ± 7.2	2.8 ± 2.9	**<0.0001**	0.651	0.290
	Preparation	12.4 ± 8.5	4.0 ± 3.2			
	Execution	11.4 ± 9.4	2.3 ± 3.0			
Early Adaptation vs. Early Retention	Control	10.5 ± 7.2	4.4 ± 3.7	**<0.0001**	0.775	0.899
	Preparation	12.4 ± 8.5	4.9 ± 3.1			
	Execution	11.4 ± 9.4	4.9 ± 3.50			
**RT (ms)**						
Baseline 1 vs. Baseline 2	Control	246.6 ± 41.6	243.3 ± 32.8	**0.010**	0.469	**0.017**
	Preparation	252.0 ± 43.8	217.0 ± 39.2			
	Execution	233.0 ± 23.7	230.5 ± 32.7			
Early Adaptation vs. Late Adaptation	Control	18.8 ± 46.28	−27.6 ± 30.4	**<0.0001**	0.600	0.216
	Preparation	17.8 ± 41.07	−2.1 ± 46.6			
	Execution	20.3 ± 35.38	−6.1 ± 58.7			
Early Adaptation vs. Early Retention	Control	18.8 ± 46.28	−14.7 ± 50.9	**<0.0004**	0.834	0.707
	Preparation	17.8 ± 41.07	−3.8 ± 31.4			
	Execution	20.3 ± 35.38	−1.4 ± 43.9			
**MT (ms)**						
Baseline 1 vs. Baseline 2	Control	305.1 ± 66.3	326.6 ± 70.0	0.063	0.894	0.106
	Preparation	311.0 ± 70.3	306.7 ± 62.9			
	Execution	299.1 ± 65.7	309.7 ± 87.1			
Early Adaptation vs. Late Adaptation	Control	146.4 ± 125.1	39.6 ± 65.6	**<0.0001**	0.651	0.290
	Preparation	152.2 ± 142.9	30.8 ± 48.1			
	Execution	192.2 ± 188.4	39.6 ± 97.8			
Early Adaptation vs. Early Retention	Control	146.4 ± 125.1	55.4 ± 110.1	**<0.0001**	0.483	0.828
	Preparation	152.2 ± 142.9	31.1 ± 76.14			
	Execution	192.2 ± 188.41	66.0 ± 75.3			

## Data Availability

The data presented in this study are available upon request from the corresponding author. The data are not publicly available due to restrictions related to ethical approval.

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
