# Peer review of "Effect of Phasic Experimental Pain Applied during Motor Preparation or Execution on Motor Performance and Adaptation in a Reaching Task: A Randomized Trial"

_brainsci, 2024, doi:10.3390/brainsci14090851_

Round 1

Reviewer 1 Report

Comments and Suggestions for Authors

Dear author, thank you for the opportunity to revise your manuscript, the following suggestion may improve the quality of your work.

Introduction

*While it introduces the concept of the impact of experimental pain on motor performance, it could more precisely delineate the specific research gap this study aims to fill.

*       The introduction would benefit from a clearer statement of the study's objectives and hypotheses. Explicitly defining these early on would enhance the reader's understanding of the study's direction.

Method

* the type of the study should be describe. You used a randomization, is the study a RCT? If yes, is the study registered in trial.gov?

*is the study approved by an institutional review board?

*The manuscript does not clearly justify the chosen sample size or discuss how it ensures sufficient power to detect expected differences between groups.

Discussion

*a more comprehensive discussion of the study's limitations would provide a more balanced view. Especially gender play an important role in pain modulation but also in muscleskeletal morphology. Plase take in to considearation in your discussion these articles concerning gender differences in muscleskeletal morphology and pain modulation:

- Deodato, M., Saponaro, S., Šimunič, B. et al. Sex-based comparison of trunk flexors and extensors functional and contractile characteristics in young gymnasts. Sport Sci Health 20, 147–155 (2024). https://doi.org/10.1007/s11332-023-01083-7

Osborne NR, Davis KD. Sex and gender differences in pain. Int Rev Neurobiol. 2022;164:277-307. doi: 10.1016/bs.irn.2022.06.013. Epub 2022 Jul 30. PMID: 36038207.

*       The discussion would benefit from more detailed suggestions for future research

Reviewer 2 Report

Comments and Suggestions for Authors

According to the authors, the article describes, in an original way, the impact of phasic experimental pain on learning and motor control in healthy subjects. The aim of this study was to evaluate the effect of experimental phasic pain occurring during the preparation or the execution of a reaching task on the acquisition and retention of sensorimotor adaptation. In order to achieve this aim, subjects were divided into three groups which enabled suitable comparisons based on the established objectives. The materials and methods used, as well as the statistical analysis, are carefully and accurately described. The results are aligned with the stated purpose and it was discussed in a consistent manner. I suggest to include the limitations as well as to add “in healthy subjects” at the end of the first sentence in the conclusion (line 450).

This article opens new perspectives of the phasic pain’s impact on other populations. This is for instance the case of older adults who experience a decrease in sensorimotor processing; subjects suffering from musculoskeletal conditions (i.e.: MTrPs); and in patients with different neurological conditions (stroke, Parkinson's) where other aspects could influence learning and motor control.

Reviewer 3 Report

Comments and Suggestions for Authors

very interesting research but it have been improved

you have a ECA so follow CONSORT statement

draw a flow chart recruitment

explain how you calaulated your sample size

present table 2 with mean and sd and p values for grou, time and time:group wilth all variables

i dont understand, you measuremente at least 4 times so you have to use an Anova across all time points, not between selected time ponts and, after, post hoc test to find between groups and time are significant differences

Round 2

Reviewer 1 Report

Comments and Suggestions for Authors

I thank authors for the revision

Reviewer 3 Report

Comments and Suggestions for Authors

very good. paper is ready now